# Greedy Output Approximation: Towards Efficient Structured Pruning for LLMs Without Retraining

Jianwei Li[1], Yijun Dong[2], Qi Lei[2]

[1]North Carolina State University, [2]New York University

jli265@ncsu.edu, yd1319@nyu.edu, ql518@nyu.edu

To remove redundant components of large language models (LLMs) without incurring significant pruning costs, this work focuses on single-shot structured pruning without a retraining phase. We simplify the pruning process for Transformer-based LLMs by identifying a depth-2 pruning structure that functions independently. Additionally, we propose two inference-aware pruning criteria derived from the optimization perspective of output approximation, which outperforms traditional training-aware metrics such as gradient and Hessian. We also introduce a two-step reconstruction technique to mitigate pruning errors without model retraining. Experimental results demonstrate that our strategy significantly reduces pruning costs and hardware requirements while maintaining superior performance across various datasets and models.

## 1. Introduction

With the development of LLMs displaying emergent capabilities like sophisticated reasoning, the focus of the community has shifted to models with billions of parameters, for example, GPT-4 and Llama2 [1, 2]. This transition introduces unprecedented computational costs both in the training and the inference phases [3–6]. To address this challenge, pruning plays a constructive role by removing redundant components from models, thereby reducing computational costs [7–10]. Notably, designing an optimal pruning strategy is an NP-hard problem (as it reduces to subset selection) and requires balancing **accuracy**, **sparsity**, **generalizability**, **pruning costs**, and **hardware compatibility** in practice [11–13]. Traditional pruning methods primarily focus on accuracy and sparsity, often neglecting other key factors. They typically involve model retraining and knowledge distillation to mitigate pruning errors. However, with current LLMs featuring billions of parameters, the training process is already a significant challenge, making the additional cost of model retraining even more unaffordable [14, 15].

Recently, some works have focused on **efficient structured pruning** on pre-trained LLMs, directly addressing hardware compatibility and generalizability. This approach allows them to concentrate on the remaining trade-off factors: sparsity, accuracy, and pruning cost. For instance, Kim et al. [16], Xia et al. [17] and Ma et al. [18] adopt single-shot pruning methods, which only require one round of retraining. In contrast, other works such as An et al. [19] and Khaki and Plataniotis [20] aim to eliminate the need for retraining completely. However, these linear approaches have respective limitations, such as high computational costs due to the use of higher-order information [12, 21, 22], a lack of fully structured pruning patterns [23–26], or compromised performance in some cases. The development of these methods highlights a critical gap in LLM pruning, namely the struggle to optimize all five factors simultaneously.

With the existing challenges, we delve into this ideal strategy by answering the following questions:

**Question 1.** *Does an uniform **pruning structure** exist in Transformer-based language models?*

We discovered depth-2 pruning modules within Transformer architecture by uniformly treating attention and feed-forward modules. These structures preserve feature knowledge while reducing pruning complexity from residual connections.

**Question 2.** *Is there effective **pruning criterion** that does not require training awareness?*

Second Conference on Parsimony and Learning (CPAL 2025).

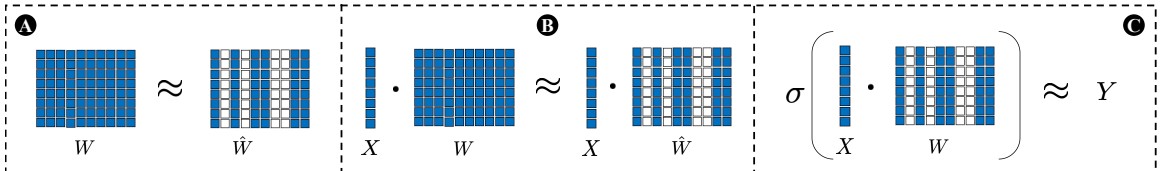

Figure 1: Pruning metric analysis from the optimization perspective **A:** Function Approximation; **B:** Output Approximation; **C:** Objective Approximation.

We identified two efficient, high-performing inference-aware pruning metrics based on output approximation for Transformer models, which are comparable to or even outperform training-aware metrics.

Answering the above questions altogether, this paper proposes an efficient, structured pruning strategy with a focus on Transformer-based LLMs [27]. Specifically, we categorize the existing pruning metrics into three groups based on their implicit purpose: function (weights) approximation, output approximation, and objective approximation (Fig. 1 describes the differences). Following the output approximation route, we introduce a similarity-based pruning strategy that exploits the redundancy in multi-head attention mechanisms by removing heads that extract similar information first rather than those with minimal impact. Additionally, we propose a second-moment-based pruning approach also under the output approximation category, which stands out for its ability to integrate information across multiple layers. We apply this metric for depth-2 modules (both attention and feed-forward modules) to remove redundant components. Finally, we develop an optimization technique that eliminates the need for higher-order information by greedily reducing pruning error through weight reconstruction of the subsequent dense module. Our structured pruning experiments on pre-trained LLMs ranging from millions to billions of parameters demonstrate that our method ensures generalizability, hardware compatibility, and minimal pruning cost. Moreover, it outperforms or achieves comparative performance to other non-retraining methods and even some methods that require retraining.

## 2. Preliminary

**Unstructured Pruning, Semi-Structured Pruning, and Fully Structured Pruning:** Unstructured pruning reduces model size by removing individual, non-essential weights, lowering storage and computational costs without significantly affecting performance [14, 28]. Semi-structured pruning introduces a specific sparsity pattern, where $N$ weights are pruned from every block of $M$, balancing flexibility with hardware efficiency. This approach imposes more structure than unstructured pruning while retaining fine-grained control [29]. Fully structured pruning, the focus of our paper, adopts a more rigid strategy by removing entire units—such as channels, neurons, or layers—making it particularly compatible with standard hardware. In contrast, unstructured and semi-structured pruning often require specialized accelerators for efficient deployment [30, 31].

**Input or Output Channel Pruning:** To clarify fully structured pruning, it is better to understand that pruning neurons can be approached in two directions: input channels and output channels. Consider a linear function $f(X) = XW$, where $X \in \mathbb{R}^{1 \times d_{in}}$ is the input and $W \in \mathbb{R}^{d_{in} \times d_{out}}$ is the weight matrix. When we prune neurons, we typically refer to pruning the output channels of $W$ since the number of neurons generally corresponds to the number of output channels in each layer. After pruning, the weight matrix becomes $\hat{W} \in \mathbb{R}^{d_{in} \times d'_{out}}$, where $d'_{out} < d_{out}$. Alternatively, pruning the input channels of $W$ equates to pruning the input $X$, also known as feature selection. This paper focuses on a static approach to feature selection, where the same channels are removed for all samples, making feature selection equivalent to output channel pruning in the previous layer. An interesting phenomenon arises: in depth-2 sequential linear layers, pruning the input channels of the second layer simultaneously pruning the output channels of the first layer, using identical pruning indices. In contrast, pruning the output channels of the second layer does not affect the first

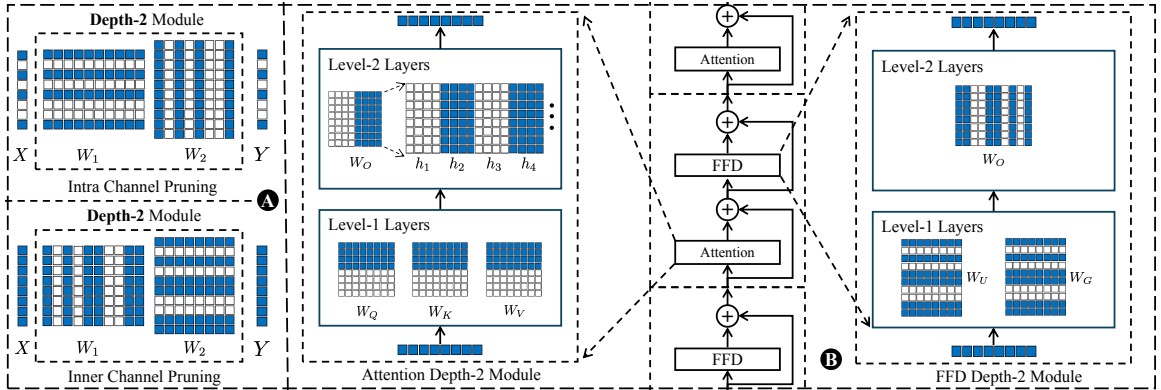

Figure 2: Pruning structure recognition. **A**: Two pruning strategies for the depth-2 module. **B**: Depth-2 modules identification in Transformer-based LLMs.

layer at all. Both of them contribute to model compression but have different potential impacts on the model performance.

**Data-free/dependent and Training/Inference-aware Pruning Metrics:** When choosing the redundant components for removing, the selection is typically guided by specific metrics [29]. These metrics can be broadly divided into data-free and data-dependent categories, depending on whether they rely on specific datasets. Additionally, they can be categorized as training-aware or inference-aware based on whether they require model backpropagation. This paper focuses on inference-aware metrics and explores both data-free and data-dependent versions.

# 3. Methodology

This section outlines our structured pruning scheme, which consists of three key components: pruning structure recognition, pruning criteria definition, and post-pruning recovery strategy.

## 3.1. Pruning Structure Recognition

Our approach involves single-shot pruning and targets structured components, such as entire rows or columns of weight matrices. We do not discuss layer or block pruning, as it disrupts inherent model correlations and requires retraining to restore layer dependencies.

### 3.1.1. Pruning Patterns in Transformer

**Depth-2 Module Identification:** In Transformer-based LLMs, both the attention and feed-forward modules operate as sequential depth-2 structures. In the attention module, the first level consists of the weight matrices $W_Q$, $W_K$, and $W_V$ (query, key, and value), which run in parallel, while the second level includes the output weight matrix $W_O$. The feed-forward module follows a similar two-level structure. A key characteristic of these depth-2 modules is that when pruning input channels at the second level, the corresponding output channel indices from the first level must also be pruned to maintain structural integrity.

**Pruning Strategies for Depth-2 Modules:** For a depth-2 module, a given compression ratio can be achieved via two pruning strategies. The first strategy involves pruning the output channels of the layers in the first level while concurrently pruning the input channels of the layers in the second level. This approach ensures that the dependencies outside the module remain invariant. The second strategy involves pruning the output channels and the initial input $X$ to the entire depth-2 module. In the context of the LLMs, which consist of multiple such modules in sequence, pruning the initial input $X$ is effectively equivalent to pruning the output channels of a preceding module in the sequence. As this dependency propagates backward through the layers, it ultimately affects the model's embedding layer, meaning we are directly pruning the channels of the token embeddings.

**Challenge of Residual Connection**: Without considering the loss of tokens' semantic information, the two pruning strategies described above should not differ significantly. However, residual connections impose substantial constraints on the second strategy. In the Transformer architecture, every depth-2 module connects a residual connection. This means that the pruned channels must be strictly aligned across all modules. If the pruned indices of one of them do not align with others, it could lead to an unpredictable loss of information. This constraint severely limits the choice of channels for pruning and could significantly decrease performance. In contrast, the first strategy maintains a fixed number of output channels across these modules, avoiding this limitation. Each module can independently select which internal channels to prune based on its needs, resulting in a larger search space for optimization. Fig. 2 describes their differences, and this paper will adopt the inner channel pruning.

**Additional Structure for Attention Mechanism**: The intricate topology of the attention block introduces an additional constraint: pruning must be conducted at the level of entire heads, encompassing continuous portions of the channels. Fortunately, given the design philosophy of multi-head attention—that each head is designed to capture correlations between tokens independently—this setup easily leads to redundancy, making it highly amenable to similarity analysis.

## 3.2. Pruning Criteria Selection

This section begins by categorizing pruning criteria based on their implicit purposes. Then, we introduce two specific pruning metrics for the aforementioned depth-2 modules and employ a magnitude-based pruning method to remove the least important channels.

### 3.2.1. Implicit Purpose of Pruning Metric

Previous work has categorized pruning metrics based on their relationship with data, as discussed in Section 5. Diverging from these approaches, we analyze these metrics based on their implicit purposes and describe them in Fig. 1. Specifically, for a linear operation $f(X) = XW$, our goal is to prune $W$ while preserving the accuracy $f(X) \approx Y$. To minimize pruning error, we can approximate $W$, $f(X)$, and $(f(X), Y)$. We term these strategies as function approximation, output approximation, and objective approximation, respectively. Function approximation focuses on directly approximating $W$, which is equivalent to approximating the function itself. Typical metrics in this category include the L1 and L2 norms of weights or neurons. Output approximation seeks to approximate the result of $XW$. Known metrics in this category include contribution energy, the sensitivity of $f(X)$ to deviations in $X$, and the variance or similarity score of $f(X)$. Objective approximation aims to directly approximate accuracy. This category encompasses metrics such as first-order or second-order information and regularization scores. However, this type of metric is computationally expensive as the optimization process involves backward propagation and calculation of the Hessian Matrix. By analyzing these strategies, this paper proposes two new metrics to guide the pruning of LLMs.

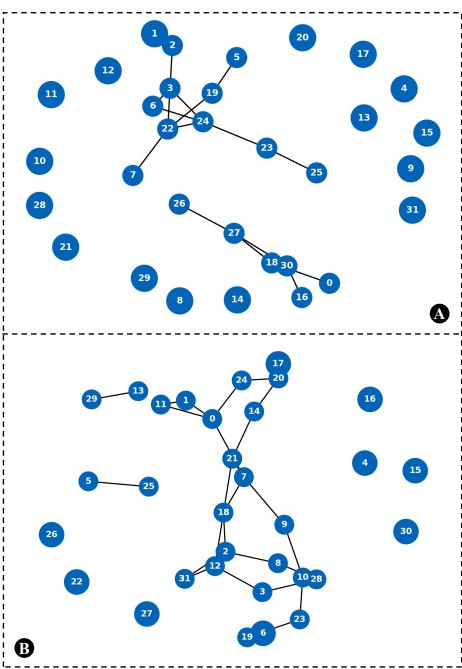

Figure 3: Similarity visualization of attention heads in **A**: block 4 and **B**: block 5 for Llama-7B. Heads with divergence less than $\tau = 0.20$ are connected.

### 3.2.2. Similarity-based Metric for Attention

Previous research on pruning attention heads typically involves removing heads with the lowest importance scores. Surprisingly, our experiments indicate that random pruning also yields competitive results compared to magnitude-based pruning, especially when the pruning ratio is below 50%.

Further experimentation with different random seeds, leading to various head indices for pruning, consistently produces comparable results. Notably, nearly all heads have been selected for removal at some point during this process, suggesting a potential oversight in our initial understanding. Recall that different attention heads are intended to capture correlations between tokens independently. Thus, it's common for similar information to be extracted across different heads. This observation prompted us to reconsider our strategy: we prioritize removing similar heads before eliminating those with the least importance score. By identifying and pruning heads that capture redundant information, we can compress the model effectively while preserving performance.

---

**Algorithm 1** Post-Pruning Recovery.

---

1: **Input:** Depth-2 module $m_i$ with $W_1$ and $W_2$; Input $X$
2: **Input:** Original dense outputs $Y_1, Y_2$ for $X$
3: **Input:** Preceding pruned modules $m_1..m_{i-1}$
4: **Output:** Reconstructed weight $\bar{W}_1$ and $\bar{W}_2$
5: **procedure** WEIGHTS RECONSTRUCTION
6: $\quad \hat{X}_1 \leftarrow (m_{i-1}(m_{i-2}..(m_1 X)))$
7: $\quad \bar{W}_1 \leftarrow (\hat{X}_1 \hat{X}_1^\top)^{-1} \hat{X}_1^\top Y_1$
8: $\quad \hat{Y}_1 \leftarrow \bar{W}_1 \hat{X}_1$
9: $\quad \hat{X}_2 \leftarrow \hat{Y}_1$
10: $\quad \bar{W}_2 \leftarrow (\hat{X}_2 \hat{X}_2^\top)^{-1} \hat{X}_2^\top Y_2$
11: **end procedure**

---

Previous studies have conducted similarity analysis between neurons [32–34], examining the output differences across multiple samples to identify similar components. The redundant neurons are then removed, and the remaining neurons scale their weights or biases to minimize the impact of this removal. However, these methods are primarily effective in smaller neural networks, as the scaling technique struggles to handle the accumulated error across numerous layers. Fortunately, due to the parallelism and independence of attention heads, removing redundant heads does not lead to significant information loss that affects subsequent layers, thus eliminating the need for costly remedial operations. Based on this observation, we define a pairwise head divergence matrix $D \in \mathbb{R}^{h \times h}$ for each attention module, where $h$ refers to the number of heads. Specifically, given an attention score matrix $Attn \in \mathbb{R}^{N \times h \times s \times s}$, where $N$ represents the number of samples and $s$ is the sequence length, let $P_i \in \mathbb{R}^{N \times s \times s}$ and $Q_j \in \mathbb{R}^{N \times s \times s}$ denote the attention scores of heads $h_i$ and $h_j$, respectively. Then $D(P_i \parallel Q_j)$ is calculated as:

$$D_{ij}(P_i \parallel Q_j) = \frac{1}{N \times s} \sum_{n=1}^{N \times s} D_{JS}(P_i^{(n)} \parallel Q_j^{(n)}) \tag{1}$$

as the empirical distance between heads $h_i$ and $h_j$ across the dataset, where $D_{JS}$ denotes the Jensen-Shannon Divergence. and $D_{ij}$ represents the empirical distance between heads $h_i$ and $h_j$ across the dataset. By visualizing the attention heads as graph nodes and connecting nodes with a divergence less than a predefined threshold $\tau$ via an edge, we can clearly illustrate the relationships between these heads. Fig. 3 demonstrates that some heads fall into the same group, signaling information redundancy, whereas others stand alone, highlighting the uniqueness of their information. We also observe that specific layers form large groups, indicating higher redundancy. The details of our pruning strategy for the attention module are outlined in Algo 2.

### 3.2.3. Second-moment-based Metric

To prune the identified depth-2 module, we follow the structure mentioned in Section 3.1, namely pruning output channels in the first level and input channels in the second level. Since the pruned channel indexes from these two directions must match, we have to consider them together. This paper proposes a second-moment-based pruning metric that is simple to calculate and incorporates information from multiple layers.

For demonstration purposes, we consider a simple linear feed-forward module $f(x) = BAx$ with weight matrices $A, B$ and Gaussian input $x \sim \mathcal{N}(0, \Sigma)$. Let $A_j$ be the $j$-th output channel (row) of $A$ such that $A_j^\top x \sim \mathcal{N}(0, A_j^\top \Sigma A_j)$; and let $B_j$ be the $j$-th column of $B$ and $B_{ij}$ be the $(i, j)$-th entry such that $Y_{ij} := B_{ij} A_j^\top x$ measures the influence of a single weight $B_{ij}$ on the output with $Y_i = \sum_j Y_{ij}$ in the $i$-th entry. The second moment of $Y_{ij}$ is given by

$$E[Y_{ij}^2] = E[B_{ij}^2 (A_j^\top x)^2] = B_{ij}^2 E[(A_j^\top x)(x^\top A_j)] = B_{ij}^2 (A_j^\top E[xx^\top] A_j) = B_{ij}^2 (A_j^\top \Sigma A_j). \tag{2}$$

Table 1: The zero-shot performance of the compressed LLaMA-7B (20% sparsity respective to the global). Following the LLM-Pruner methodology [18], we only prune the transformer blocks from the 4th to the 30th. The average performance is calculated across seven classification datasets. **Bold** indicates the best pruning-only performance, while underline represents the **overall-best** performance.

| Pruning Methods | WikiText2↓ | PTB↓ | BoolQ | PIQA | HellaSwag | WinoGrande | ARC-e | ARC-c | OBQA | Ave↑ |
|---|---|---|---|---|---|---|---|---|---|---|
| Dense | 12.62 | 22.14 | 73.18 | 78.35 | 72.99 | 67.01 | 67.45 | 41.38 | 42.4 | 63.5 |
| *Data Free Pruning* | | | | | | | | | | |
| Random | 23.02 | 40.19 | 46.21 | 71.33 | 59.35 | 56.51 | 47.97 | 32.0 | 36.30 | 49.95 |
| L1 norm | 179.02 | 311.75 | 51.28 | 60.22 | 43.14 | 52.01 | 36.53 | 27.89 | 30.8 | 43.12 |
| L2 norm | 582.41 | 1022.17 | 60.18 | 58.54 | 37.04 | 53.27 | 32.91 | 27.56 | 29.8 | 42.76 |
| Ours (Self-Gen) | 21.76 | 34.3 | 63.51 | 72.63 | 56.54 | 54.46 | 51.68 | 33.79 | 36.4 | 52.72 |
| *Data Dependent Pruning* | | | | | | | | | | |
| *Training-Aware Pruning Criterion* | | | | | | | | | | |
| LLM-Pruner Vec | 22.28 | 41.78 | 61.44 | 71.71 | 57.27 | 54.22 | 55.77 | 33.96 | 38.4 | 53.52 |
| LLM-Pruner E1 | 19.09 | 34.21 | 57.06 | 75.68 | 66.8 | 59.83 | 60.94 | 36.52 | 40.0 | 56.69 |
| LLM-Pruner E2 | 19.77 | 36.66 | 59.39 | 75.57 | 65.34 | 61.33 | 59.18 | 37.12 | 39.8 | 56.82 |
| *Inference-Aware Pruning Criterion* | | | | | | | | | | |
| Wanda-sp | 27.45 | 49.52 | 64.16 | 75.21 | **68.62** | 62.27 | 59.68 | 36.68 | 39.2 | 57.97 |
| Ours (Calibration) | **17.48** | 30.04 | 66.48 | 75.78 | 67.73 | 62.27 | 61.4 | 35.49 | 39.6 | 58.39 |
| Ours C w/ remedy | 17.90 | 31.23 | 70.12 | **76.86** | 68.55 | 65.76 | **64.23** | **38.54** | **40.5** | **60.65** |
| *Training-Aware Pruning Criterion and Model Retraining* | | | | | | | | | | |
| LLM-Pruner LoRA | 17.37 | 30.39 | 69.54 | 76.44 | 68.11 | 65.11 | 63.43 | 37.88 | 40.0 | 60.07 |

To obtain an importance score for the $j$-th inner channel $B_j$ based on the second moment, we take the sum over all output channels in $B$:

$$\mathcal{M}_j := \sum_i E[Y_{ij}^2] = \|B_i\|_2^2 (A_j^\top \Sigma A_j) \tag{3}$$

For a depth-2 module $f(x) = B\sigma(Ax)$ with a more practical ReLU activation $\sigma$, we scale the above equation with a constant factor of $\frac{1}{2}$ to accommodate for the reduction in variance when $A_j^\top x < 0$. Further extending the above notion of second-moment-based importance score to GeLU or SiLU activations, we empirically observe that the additional nonlinearity in $\sigma$ has negligible influence on $Y_{ij}$ and therefore treat these nonlinear activations the same as ReLU, i.e.

$$\mathcal{M}_j := \frac{1}{2}\|B_i\|_2^2 (A_j^\top \Sigma A_j). \tag{4}$$

This approach offers more valuable information from the covariance matrix compared to methods based on output energy (first moment) [35–37]. Unlike some statistical methods that require calibration datasets to collect feature values and then calculate statistical properties, our method can flexibly integrate information from both input and output channels, whereas those methods are limited to focusing only on output channels. When there is no prior information on $\Sigma$, we assume isotropic Gaussian data with $\Sigma$ being the identity. More details about calculating $\mathcal{M}_i$ can be found in the Appendix 6.

## 3.3. Recovery Without Retraining

With the selection of the pruning structure and criteria, this paper proposes a module-wise pruning approach. Similar to layer-wise pruning, we prune these depth-2 modules sequentially. Notably, due to errors introduced by pruning preceding modules, the input to the current module inevitably deviates from its dense version. Consequently, even without pruning the current module, a discrepancy between its output and the original output is unavoidable. Recall that our design philosophy is to approximate the output as closely as possible. Thus, it is crucial to reconstruct the weights of the current module before pruning. This paper presents a reconstruction technique in Algo. 1, 3, 4 that can mitigate the accumulated errors without requiring model retraining. This reconstruction process ensures that the output of each module can still align as closely as possible with the original, even with the new input. This way, the pruning criteria for each channel can be optimally up-to-date.

Unlike Li et al. [12], which primarily focuses on reconstructing a single layer, our approach targets more complex structures, including intricate layer dependencies.

Table 2: Perplexity of compressed GPT-2 and LLaMA-7B (25% and 50% sparsity respective to pruned blocks) on Wikitext2 and PTB. We prune the 4th to 30th transformer blocks for LLaMA-7B and all blocks for GPT-2. **Bold** indicates the best performance, while underline represents the **second-best** performance.

| Models | GPT-2: [0-12) | | | | LLama-7b: [4-30) | | | |
|---|---|---|---|---|---|---|---|---|
| Datasets: PPL | WikiText2: 29.95↓ | | PTB: 40.12↓ | | WikiText2: 12.62↓ | | PTB: 22.14↓ | |
| Sparsity | 25% | 50% | 25% | 50% | 25% | 50% | 25% | 50% |
| Data Free Pruning | | | | | | | | |
| Random | 189.73 | 1839.33 | 245.33 | 2769.6 | 23.02 | 100.42 | 40.19 | 133.56 |
| L1 norm | 338.3 | 1226.13 | 583.2 | 1290.45 | 179.02 | 891.23 | 311.75 | 1034.69 |
| L2 norm | 227.32 | 674.52 | 324.33 | 800.14 | 582.41 | 14000.68 | 1022.17 | 28062.45 |
| Ours (Self-Gen) | **119.29** | **586.87** | **152.93** | **723.39** | **21.76** | **58.61** | **34.3** | **64.24** |
| Data Dependent Pruning | | | | | | | | |
| Training-Aware Pruning Criterion | | | | | | | | |
| LLM-Pruner Element1 | 9229.32 | 32453.23 | 11993.24 | 8020.87 | 19.09 | 48.84 | 34.21 | 105.24 |
| LLM-Pruner Element2 | 1897.32 | 14706.23 | 2258.33 | 18598.33 | 19.77 | 72.89 | 36.66 | 138.33 |
| LLM-Pruner Vector | 488.32 | 39025.12 | 6169.56 | 18616.87 | 22.88 | 55.68 | 41.76 | 305.24 |
| Inference-Aware Pruning Criterion | | | | | | | | |
| Wanda-Structured Pruning | 586.34 | 4147.32 | 355.17 | 3246.79 | 27.45 | 69.02 | 49.52 | 132.52 |
| FLAP UL-UM w/o remedy | 818.14 | 3636.23 | 554.32 | 2758.37 | 17.15 | 36.08 | 34.96 | 85.22 |
| FLAP UL-UM w/ remedy | 2197.32 | 3043.35 | 2199.24 | 3561.76 | 15.76 | 26.87 | 32.1 | 66.18 |
| Ours (Calibration) UL-UM | 81.96 | 317.37 | 186.68 | 936.57 | NA | NA | NA | NA |
| FLAP AL-AM w/o remedy | 126.57 | 5538.32 | 135.07 | 10244.95 | 17.01 | 34.09 | 30.99 | 71.76 |
| FLAP AL-AM w/ remedy | 1349.25 | 5382.14 | 1769.56 | 7476.08 | **15.06** | **26.55** | **29.45** | 57.89 |
| Ours (Calibration) ML-MM | **79.4** | **251.34** | **130.54** | **756.33** | 17.48 | 26.87 | 30.04 | 57.89 |

# 4. Experiment

This section initially presents the fundamental setup for our experiments. Subsequently, we demonstrate and analyze the results from multiple perspectives.

**Baselines:** This paper compares state-of-the-art pruning methods across multiple dimensions, aiming for fair evaluations and in-depth analyses to uncover the reasons behind the observed results. First, we compare our approach with data-free pruning methods, including **random** pruning and **magnitude-based** pruning (**L1** and **L2** norms) [29]. Next, we evaluate our methods against data-dependent pruning techniques, encompassing training-aware, inference-aware, and retraining-required methods. In the training-aware category, we compare with various configurations of **LLM-Pruner** [18], such as **Element1**, **Element2**, and **Vector-wise** magnitude pruning. Within the inference-aware category, we compare with the structured version of **Wanda** [24] and **FLAP** [19]. Additionally, we extend our comparisons to include the LLM-Pruner method augmented with one-round **LoRA** retraining Hu et al. [38]. Such comprehensive evaluations will demonstrate the effectiveness of our pruning approach.

**Models:** Our primary experiments are categorized into two series based on the model scale: **LLaMA-7B** with 7 billion parameters and **GPT-2** with 110 million parameters [3, 39]. This aligns with our study's goal to assess pruning performance across different model sizes and ensure a thorough examination. Additionally, we extend our experiments to other models, including **Vicuna-7B** [40], **Llama2-7B** [2], and **Llama3.1-8B** [41]. This comprehensive selection allows us to explore a broader spectrum of capabilities and sizes, enhancing our understanding of how different architectures perform under various computational constraints. Additional experiment results can be found in Appendix 6.

**Evaluation and Datasets:** To evaluate performance, we adopt LLaMa's approach by conducting zero-shot task classification on a range of common sense reasoning datasets: **BoolQ** [42], **PIQA** [43],

**HellaSwag** [44], **WinoGrande** [45], **ARC-easy** [46], **ARC-challenge** [46], and **OpenbookQA** [47]. Following the methodology in [48], the model either ranks the options in multiple-choice tasks or generates answers in open-ended formats. Additionally, we follow Ma et al. [18] to conduct a zero-shot perplexity (PPL) analysis on **WikiText2** [49] and the **Penn Treebank** (PTB) [50] with a specific sequence length 128.

**Implementation:** During the pruning phase, we randomly select 16 samples from **Wikitext2** or **Bookcorpus** [51], truncated to a sequence length of **128** for **LLaMA-7B** and **1024** for **GPT-2**. These samples serve as calibration data for pruning metric calculation and covariance matrix extraction, respectively. During the recovery phase, we sample an additional 1024 examples from the downstream dataset to guide optimization in the data-dependent comparison experiments.

## 4.1. Results and Analysis

We present the main results in Tab. 1. For the data-free comparison experiments, we leverage the inherent ability of LLMs to generate sentences. Our pruning method uses these generated sentences as calibration data because, given that the LLMs are well-trained, these sentences naturally conform to the semantic and syntactic token distributions of the training data. Compared to traditional data-free metrics (L1 or L2), our data-free version, which relies solely on the model itself, achieves significant improvements in perplexity and up to a 20% enhancement in zero-shot evaluation for downstream tasks. Moreover, our method surpasses random pruning by at least 6%, a significant improvement achieved without relying on existing datasets, while traditional metrics (L1 or L2) fail

Table 3: Similarity-based analysis for LLaMA-7B attention heads pruning (all blocks) with different $\tau$. 'Bold' indicates the best performance.

| Methods | # pruned heads | Wiki2 ↓ | PTB ↓ |
|---|---|---|---|
| Dense | 0 | 12.62 | 22.14 |
| Ours ($\tau = 0.16$) | 88 | **12.96** | **22.45** |
| Random | 64 | 14.50 | 24.13 |
| L2 Norm | 64 | 14.69 | 25.64 |
| 1st+2nd order | 64 | 13.45 | 24.19 |
| FLAP | 88 | 12.90 | 22.67 |
| Ours ($\tau = 0.19$) | 204 | 14.69 | **24.32** |
| Random | 192 | 18.75 | 35.73 |
| L2 Norm | 192 | 195.84 | 371.65 |
| 1st+2nd order | 192 | 14.81 | 28.77 |
| FLAP | 204 | **13.22** | 24.42 |

to outperform. These results demonstrate the superiority of our techniques in data-free pruning methods.

Our approach outperforms data-dependent pruning methods and the inference-only method Wanda-SP. Impressively, it also surpasses the state-of-the-art training-aware pruning method LLM-Pruner, which includes different configurations such as Element1, Element2, and Vector. Our approach consistently demonstrates better pruning results without requiring computationally intensive first-order and second-order information. Moreover, our method even achieves better results compared to LLM-Pruner with LoRA, despite the latter involving model retraining.

We also compare our method with the state-of-the-art inference-only method FLAP and present the results in Tab. 2. Our approach exhibits significantly better results on the GPT-2 model and achieves comparable performance with LLaMA-7B. Overall, our method demonstrates superior performance in both data-free and data-dependent pruning categories.

## 4.2. Ablation Study

We also explore our pruning metrics by exclusively pruning attention heads. The experimental results in Tab. 3 demonstrate that for colossal LLMs like LLaMA-7B, our similarity analysis effectively identifies redundant attention heads with minimal negative impact on model performance. Compared to inference-aware metrics such as the L2 norm, training-aware metrics using first- and second-order information, and random pruning, our similarity-based metric consistently outperforms. When compared to the specifically designed metric of FLAP, we achieve better or comparable performance. These results strongly indicate that we should prioritize pruning redundant information rather than heads with small importance scores.

Additionally, we designed experiments to explore the influence of the number of calibration samples. Figure 4 shows that in LLaMA-7B pruning-only experiments, our method is insensitive to the number of calibration samples, achieving comparable results with as few as 8 samples and as many as 128

samples. Conversely, in GPT-2 pruning with remediation experiments, performance improves with an increasing number of calibration samples. These findings demonstrate that our pruning method is robust regardless of the number of calibration samples, while our pre-pruning recovery method benefits from a higher number of calibration samples. However, this improvement gradually diminishes once the number of samples reaches a threshold.

## 5. Related Work

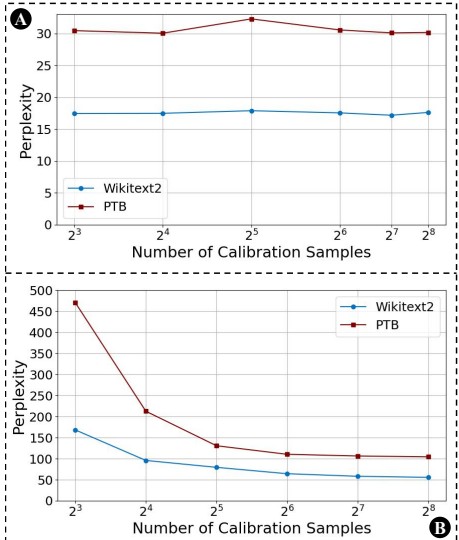

Figure 4: Performance of compressed **A**: LLaMA-7B (w/o Remediation) and **B**: GPT-2 (w/ Remediation) concerning the number of calibration samples.

**Efficient and Low-Resource Pruning:** With the growing parameter size in LLMs, efficient pruning has become essential. Methods like LLM-Pruner, Sheared LLaMA, and Shortened LLaMA use single-shot pruning but require retraining and rely on costly metrics [16–18]. In contrast, approaches like OPTIN and SlicedGPT eliminate retraining but still depend on computationally expensive second-order Hessian information [20–22, 24]. Meanwhile, FLAP and Wanda design specific pruning metrics inspired by pre-deep learning era methods [32–34, 52], significantly reducing computational demand [19, 26]. This paper presents a method that avoids both retraining and costly metrics while delivering superior or comparable performance to others. In parallel with this paper, [53, 54] also focus on static structured pruning without requiring model retraining, [55] find a way to break the residual dependency issue of intra-channel pruning.

## 6. Discussion and Conclusion

**Implicit Motivation and Call:** In the pre-deep learning era, various pruning metrics and structures were designed. For example, variance-based pruning and bias-based remedy methods similar to FLAP were proposed by researchers 30 years ago [32–34, 52]. These early researchers already recognized that feature information is at least as crucial as model weights in constructing pruning criteria. In the early stages of deep learning (before 2022), many researchers found that multi-round model retraining could easily recover the lost performance induced by pruning, even when based solely on weight magnitudes. As a result, the importance of pruning metrics and structure design was often overlooked, with reliance placed on retraining to validate methods. However, this paradigm shifted after 2022, when colossal LLMs became mainstream in the community. Training such models is prohibitively expensive, making pruning that relies on multi-round retraining impractical. Although parameter-efficient training methods like LoRA can reduce costs, they still require rigorous data selection [18, 38]. Thus, we urge the community to return to designing metrics that better account for the influence of both weights and features, rather than focusing solely on dataset competition. Motivated by this, this paper focuses on inference-aware pruning metrics that do not require retraining.

**Limitation:** This work evaluates the compressed LLMs primarily on perplexity and downstream tasks. However, we do not assess the emergent abilities of colossal LLMs, such as mathematical reasoning, safety alignment, common sense reasoning, contextual understanding, and creativity in text generation. Future research will focus on evaluating and enhancing these emergent abilities to provide a more comprehensive understanding of the compressed LLMs.

**Conclusion:** This paper introduces a novel approach to pruning LLMs by identifying a depth-2 pruning structure and developing two inference-aware pruning criteria. These strategies surpass traditional metrics and eliminate the need for computationally expensive retraining. Our two-step reconstruction technique further mitigates the pruning error, ensuring superior performance across

various datasets. Overall, our approach reduces pruning costs and hardware requirements, offering an efficient solution for LLM pruning.

## Acknowledgement

The authors wish to thank the anonymous reviewers for their helpful comments. The authors also would like to extend their sincere gratitude to the ARC (A Root Cluster for Research into Scalable Computer Systems) at the Computer Science Department of North Carolina State University. The invaluable computing resources provided by the ARC cluster (https://arcb.csc.ncsu.edu/ mueller/cluster/arc/) were instrumental in facilitating the research presented in this paper.

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

# Appendix-A: Pruning in LLMs Era

Pruning is a promising method that can effectively reduce model inference costs. In the main part of this paper, we discuss pruning methods within different classification philosophies. We summarize previous work and categorize pruning from multiple perspectives: **unstructured** and **semi/fully structured**, **data-free** and **data-dependent**, **training-aware** and **inference-aware**, and **retraining-free** and **retraining-dependent**. We also propose an innovative **optimization-oriented** view of pruning, which involves: **A:** Function Approximation, **B:** Output Approximation, and **C:** Objective Approximation. Our pruning pattern is designed based on **B**.

Additionally, we review more about pruning, including the most popular techniques in the pre-LLMs era (before 2022), such as **Iterative Magnitude Pruning** and the comparison between **Randomized Pruning** and **Magnitude-based Pruning**. We also discussed the relationship between **Pruning** and **Quantization**. By considering these various dimensions and methodologies, we aim to provide a comprehensive understanding of pruning and its potential to enhance model efficiency **in the age of LLMs**.

## Iterative Magnitude Pruning in Pre-LLMs Era

Iterative Magnitude Pruning (IMP) is the most renowned strategy for achieving state-of-the-art results, surpassing other methods such as Single-shot Network Pruning (SNIP) [14, 28, 56, 57]. This approach divides the pruning process into multiple stages by gradually increasing the sparsity. At each stage, the goal is to identify and remove redundant parameters or neurons. The most intuitive approach is to assign an importance score to each element and keep only the top-k elements, where the score can be based on the absolute value of weights, output sensitivity, gradients, or other fine-designed metrics [58–64]. Weight magnitude is the most straightforward and data-free method, while other metrics can be computationally expensive as they require training with data [65–70]. Moreover, IMP is accompanied by a retraining phase to restore performance, which can be computationally costly. Therefore, in the era of colossal LLMs, IMP and other methods that heavily depend on model retraining are no longer effective due to the immense costs involved.

---

**Algorithm 2** Attention Heads Pruning.

---

1: **Input:** Pairwise head divergence matrix $D \in \mathbb{R}^{h \times h}$
2: **Input:** divergence threshold $\tau$
3: **Output:** List of candidate heads for pruning $C$
4: Initialize $C = []$
5: **for** row $i$ **and** col $j$ **in** $D$ **do**
6:     **if** $D[i][j] < \tau$ **and** $i \neq j$ **then**
7:         **if** $i \notin C$ **and** $j \notin C$ **then**
8:             $C.append(i)$
9:         **end if**
10:     **end if**
11: **end for**
12: **Prune** $C$

---

## Randomized Pruning v.s. Magnitude Pruning in LLMs Era

Excluding the influence of model retraining, we discovered an interesting phenomenon for model pruning. For colossal LLMs such as LLaMA-7B, randomized pruning surprisingly produced competitive results. Specifically, compared to traditional data-free pruning metrics like L1 and L2 norm values, randomized pruning achieved several times better results, even rivaling data-dependent pruning methods. However, this advantage only existed when the pruning ratio was less than 2x. As the pruning ratio increased, magnitude pruning gradually yielded better results. Initially, we attributed this phenomenon to the high redundancy of parameters in LLMs. However, our experiments with GPT-2 showed that randomized pruning was still weaker than magnitude pruning.

**Therefore, we speculate that for colossal LLMs like Llama-7B, feature plays a more crucial role than weights in model pruning compared to smaller LLMs like GPT-2.**

Magnitude-based pruning methods aim to remove weights or neurons from a neural network that appear least influential, primarily determined by the value of their weights. The rationale behind these methods is to reduce overall model size and computational requirements without a drastic loss in performance. However, several challenges arise with this approach, and one major challenge is the lack of variety if the magnitude is based on data-free metrics (L1 or L2). This kind of metric focuses solely on the magnitude of the weights for pruning decisions, potentially missing smaller weights that play pivotal roles, especially in edge cases or rarer instances. To illustrate this more clearly, consider the following example. The output of a neural network can be represented as $y = \sum(w_i \cdot f_i)$, where $y$ is the network output, $f_i$ represents a feature, and $w_i$ is the corresponding weight. In magnitude-based pruning (L1 or L2), if $|w_i| < \tau$ ($\tau$ is pruning threshold), then $w_i \cdot f_i$ is pruned. However, the impact on $y$ is not solely determined by $w_i$, but by the combined effect of $w_i$ and the sensitivity of $f_i$. For instance, if $f_i$ represents the sharpness of an image, even a small weight $|w_i| = 0.01$ can significantly affect $y$ if $f_i$ is highly sensitive, such as affecting object recognition. Conversely, if $f_i$ represents the hue of an image background, a large weight $w_i = 5$ might have minimal impact on $y$ if $f_i$ is less sensitive, such as the background hue not altering recognition much. The influence on $y$ is thus a joint effect of $w_i$ and the sensitivity of $f_i$. This example indicates that the influence of feature information plays a significant role in identifying redundant elements.

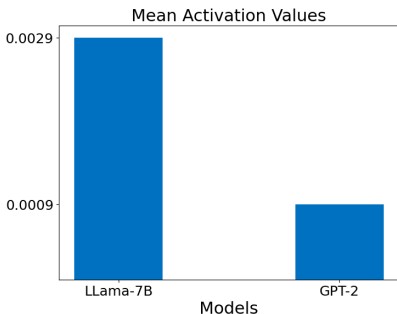

Figure 5: Mean activation value Llama-7B and GPT-2 on Wikitext2.

Based on the above observation, we speculate that LLaMA-7B's feature information contributes more to the importance score of removed elements when the pruning ratio is less than 2x. As the pruning ratio gradually increases, the influence of the features on the activation values is no longer greater than that of the weights. Therefore, randomized pruning fails at larger pruning ratios. To validate our hypothesis, we conducted a statistical analysis on the feature values of LLaMA-7B, described in Figure 5. Our results show that colossal LLMs like Llama-7B have larger activation values than smaller LLMs like GPT-2. **These findings further motivate us to design the pruning metrics that incorporate both feature and weight information instead of seeking dataset competition.**

### Pruning v.s. Quantization in LLMs Era:

Pruning, though considered less effective than quantization in the era of colossal LLMs, should not be underestimated. In practice, pruning and quantization can complement each other, yielding significant benefits when applied together [14]. Even pruning a small percentage of parameters, such as 5%, can be valuable if it meets practical performance requirements. Therefore, integrating pruning into the optimization process is always worthwhile.

## Appendix-B: More Details of 2nd-Moment-Based Metric

In Section 3.1.1, we introduced the 2nd-moment-based pruning metric for a standard depth-2 module. However, there are different variants of depth-2 modules, including the attention module and the

gated feed-forward module. We describe the metric calculation for these variants in the following section.

**Notations:** To better demonstrate our method, let us first establish the notations. We focus on the pruning of Transformer-based large language models, thus we refer to the attention mechanism as $\mathbf{Cat}_{i=1}^{h}[\sigma_1(\mathbf{XW}_i^K\mathbf{W}_i^Q\mathbf{X}^\top)\mathbf{XW}_i^V]\mathbf{W}^O$, with $i$ indicating the attention head index. The symbols $W^K$, $W^Q$, $W^V$ and $W^O$ represent the weights for the key, query, value, and output in the attention block, respectively. For the general and gated feed-forward module, we denote the logic as $\mathbf{W}^D\sigma_2(\mathbf{W}^U\mathbf{X})$ and $\mathbf{W}^D(\mathbf{W}^U\mathbf{X} \cdot \sigma_2(\mathbf{W}^G\mathbf{X}))$. Here, $W^U$, $W^D$, and $W^G$ stand for the weights for upward projection, downward projection, and gate projection. $\sigma$ refers to the activation function for all of them: SoftMax, ReLU, GeLU, or SiLU function.

Based on the above notations, we can treat the entire output of $\sigma_1(\mathbf{XW}i^K\mathbf{W}i^Q\mathbf{X}^\top)\mathbf{X}$ as the input to $\mathbf{W}_i^V$. Let $\hat{X}$ represent this new input. In this way, the attention module can be viewed as a module similar to a standard depth-2 module $(\hat{X}\mathbf{W}_i^V)\mathbf{W}_i^O$, with each level having only one linear layer. Notely, we need to view an attention module as $m$ standard depth-2 modules ($m$ is the number of attention heads), as the attention heads operate independently. For the gated feed-forward module $\mathbf{W}^D(\mathbf{W}^U\mathbf{X} \cdot \sigma_2(\mathbf{W}^G\mathbf{X}))$, we treat it as a product of two standard depth-2 modules. Specifically, we can divide it into two modules: $\mathbf{W}^D \cdot \mathbf{W}^U\mathbf{X}$ and $\mathbf{W}^D(\sigma_2(\mathbf{W}^G\mathbf{X}))$, and calculate the 2nd-moment metric separately for them. Finally, we use the product of their own metric as the 2nd-moment metric for the entire module. These approaches allow us to effectively prune channels in attention and gated feedforward modules by leveraging the 2nd-moment-based metric.

# Appendix-C: More Details of Post-Pruning Recovery Method

In Section 3.3, we only provide the post-pruning recovery algorithm for the standard depth-2 module, thus we describe the details of the recovery process for the attention module and gated feed-forward module in Algo 3 and Algo 4, respectively. All of them include multiple linear layers in the first level of the depth-2 module.

---

**Algorithm 3** Post-Pruning Recovery for Attention Module.

---

1: **Input:** Attention module layers with weights $W_K, W_Q, W_V, W_O$
2: **Input:** Corresponding inputs $X$
3: **Input:** Corresponding outputs $Y_K, Y_Q, Y_V, Y_O$
4: **Input:** Function reconstruct_best_weight and restore_layer_weights
5: **Output:** Reconstructed weights $\bar{W}_K, \bar{W}_Q, \bar{W}_W, \bar{W}_O$
6: **procedure** WEIGHTS RECONSTRUCTION
7:     $XTX \leftarrow \text{Matmul}(X.T, X)$
8:     $\bar{W}_K \leftarrow \text{reconstruct\_best\_weight}(XTX, X, Y_K)$
9:     $\bar{W}_Q \leftarrow \text{reconstruct\_best\_weight}(XTX, X, Y_Q))$
10:     $\bar{W}_V \leftarrow \text{reconstruct\_best\_weight}(XTX, X, Y_V))$
11:     $\text{restore\_layer\_weights}(module.k\_proj, \bar{W}_{k\_proj})$
12:     $\text{restore\_layer\_weights}(module.q\_proj, \bar{W}_{q\_proj})$
13:     $\text{restore\_layer\_weights}(module.v\_proj, \bar{W}_{v\_proj})$
14:     - - - - - - - - - - - - - - - - - - - - - - - - - - - -
15:     $X_2 \leftarrow module(X)$ # new input for o proj
16:     $XTX_2 \leftarrow \text{Matmul}(X_2.T, X_2)$
17:     $\bar{W}_O \leftarrow \text{reconstruct\_best\_weight}(XTX_2, X_2, Y_O)$
18:     $\text{restore\_layer\_weights}(module.o\_proj, \bar{W}_O)$
19: **end procedure**

---

# Appendix-D: More Implementation Details and Experiment Results

In this paper, we evaluated our pruning method on LLaMA-7B and GPT-2 models with a sequence length of 128 tokens for LLaMA-7B and 1024 tokens for GPT-2. This setup was chosen to ensure

consistent with other structured pruning baselines, such as LLM-Pruner and FLAP, and to accommodate hardware constraints. Given that LLaMA-7B is a large-scale model, evaluating with a shorter sequence length (128 tokens) allowed us to reduce computational overhead, making the experiments feasible on general-purpose GPUs like NVIDIA A6000. By using a specific sequence length of 128, we aimed to maintain consistency with structured pruning setups and ensure a fair comparison with existing works.

For the data-free comparison experiments, instead of using an identity matrix to calculate the 2nd-moment-based importance score, we extract the covariance matrix for the depth-2 modules using random input. Specifically, we generate random token IDs from the model's vocabulary to simulate input data. It is important to note that the self-generated calibration data does not participate in this extraction process.

---

**Algorithm 4** Post-Pruning Recovery for Gated Feed-forward Module.

---

  1: **Input:** Gated feed-forward module with weights $W_U, W_G, W_D$
  2: **Input:** Corresponding inputs $X$
  3: **Input:** Corresponding outputs $Y_U, Y_G, Y_D$
  4: **Input:** Function reconstruct_best_weight and restore_layer_weights
  5: **Output:** Reconstructed weights $\{\bar{W}_U, \bar{W}_G, \bar{W}_D\}$
  6: **procedure** WEIGHTS RECONSTRUCTION
  7:     $XTX \leftarrow \text{Matmul}(X.T, X)$
  8:     $\bar{W}_U \leftarrow \text{reconstruct\_best\_weight}(XTX, X, Y_U)$
  9:     $\bar{W}_G \leftarrow \text{reconstruct\_best\_weight}(XTX, X, Y_G))$
10:     restore_layer_weights(module.up_proj, $\bar{W}_U$)
11:     restore_layer_weights(module.gate_proj, $\bar{W}_G$)
12:     - - - - - - - - - - - - - - - - - - - - - - - - - - - - -
13:     $X_2 \leftarrow \text{module}(X)$ # new input for down proj
14:     $XTX_2 \leftarrow \text{Matmul}(X_2.T, X_2)$
15:     $\bar{W}_D \leftarrow \text{reconstruct\_best\_weight}(XTX_2, X_2, Y_D)$
16:     restore_layer_weights(module.down_proj, $\bar{W}_D$)
17: **end procedure**

---

To provide a more comprehensive evaluation of our methods, we conducted additional pruning experiments on various large language models (LLMs), including Vicuna-7B, LLaMA2-7B, and LLaMA3.1-8B. The results of these experiments are presented in Tables 6, 4, and 5. As the experimental data shows, our method consistently outperforms various baselines in both data-free and data-dependent pruning settings. Our approach is inference-aware and performs as well as training-aware metrics, which typically integrate both first-order and second-order information. However, we did not replicate baseline methods that require retraining on LLaMA2-B or LLaMA3.1-B models, as we aimed to eliminate the influence of hyperparameter tuning. Interestingly, we observed that in some cases, such as with LLaMA3.1, our data-free pruning approach even outperformed data-dependent pruning in downstream tasks, despite perplexity analysis indicating a different trend. This suggests that downstream task performance does not always align with perplexity metrics.

## Appendix-E: Broader Impact

Pruning large language models (LLMs) has significant implications for the environment and the accessibility of advanced AI technologies. By reducing these models' size and computational demands, pruning techniques can lower the energy consumption and carbon footprint associated with training and deploying LLMs, contributing to more sustainable AI practices. Additionally, the ability to run more efficient models on lower-cost hardware democratizes access to cutting-edge AI tools, enabling wider participation from researchers, developers, and institutions with limited resources. However, there are potential risks to consider, such as the possibility that pruned models might exacerbate biases or lose important contextual knowledge if not carefully evaluated. Thus, it is essential to assess not only the performance but also the ethical and societal impacts of these techniques, ensuring that they contribute positively to the broader AI landscape.

Table 4: The zero-shot performance of the compressed Llama2-7B (20% sparsity respective to the global). Following the LLM-Pruner methodology [18], we only prune the transformer blocks from the 4th to the 30th. The average performance is calculated across seven classification datasets.

| Pruning Methods | WikiText2 ↓ | PTB ↓ | BoolQ | PIQA | HellaSwag | WinoGrande | ARC-e | ARC-c | OBQA | Ave ↑ |
|---|---|---|---|---|---|---|---|---|---|---|
| Dense | 12.2 | 48.4 | 71.1 | 78.4 | 72.9 | 67.3 | 69.3 | 40.6 | 40.8 | 62.9 |
| Data Free Pruning | | | | | | | | | | |
| Random | 25.7 | 87.9 | 61.7 | 70.4 | 56.6 | 58.8 | 54.7 | 30.2 | 35.0 | 52.5 |
| L2 norm | 895.9 | 1540.2 | 53.4 | 55.7 | 32.3 | 51.4 | 29.7 | 29.8 | 31.4 | 40.5 |
| L1 norm | 239.9 | 1038.2 | 55.9 | 58.9 | 39.9 | 49.9 | 32.1 | 29.3 | 29.0 | 42.1 |
| Ours SG w/o remedy | **21.0** | **74.0** | 63.5 | 73.2 | 66.7 | 63.0 | 60.8 | 35.0 | 37.6 | **57.1** |
| Data Dependent Pruning | | | | | | | | | | |
| Training-Aware Pruning Criterion | | | | | | | | | | |
| LLM-Pruner Vec | 20.52 | 73.76 | 59.6 | 73.8 | 62.3 | 62.8 | 59.6 | 35.2 | 37.0 | 55.8 |
| LLM-Pruner E1 | 19.2 | 72.61 | 62.2 | 76.2 | 65.8 | 60.7 | 63.8 | 37.2 | 39.0 | 57.8 |
| LLM-Pruner E2 | 19.09 | 72.89 | 48.8 | 76.0 | 65.7 | 60.0 | 65.0 | 37.5 | 39.0 | 56.0 |
| Inference-Aware Pruning Criterion | | | | | | | | | | |
| Ours (C) w/o remedy | **17.4** | **64.1** | 66.4 | 74.0 | 66.0 | 63.4 | 61.7 | 35.4 | 39.0 | **58.0** |

Table 5: The zero-shot performance of the compressed Llama3.1-8B (20% sparsity respective to the global). Following the LLM-Pruner methodology [18], we only prune the transformer blocks from the 4th to the 30th. The average performance is calculated across seven classification datasets.

| Pruning Methods | WikiText2 ↓ | PTB ↓ | BoolQ | PIQA | HellaSwag | WinoGrande | ARC-e | ARC-c | OBQA | Ave ↑ |
|---|---|---|---|---|---|---|---|---|---|---|
| Dense | 14.3 | 27.7 | 82.1 | 81.1 | 73.6 | 72.5 | 81.5 | 53.6 | 45.0 | 69.9 |
| Data Free Pruning | | | | | | | | | | |
| Random | 45.8 | 76.4 | 61.6 | 71.5 | 56.5 | 59.4 | 58.9 | 34.6 | 33.4 | 53.7 |
| L2 norm | 88.5 | 149.3 | 57.5 | 68.1 | 55.9 | 57.1 | 48.1 | 32.8 | 36.0 | 50.8 |
| L1 norm | 230.9 | 376.3 | 53.2 | 61.8 | 39.3 | 53.2 | 37.5 | 25.5 | 29.4 | 42.8 |
| Ours SG w/o remedy | **35.3** | **61.6** | 68.8 | 75.4 | 64.5 | 64.0 | 66.6 | 39.8 | 40.8 | **59.9** |
| Data Dependent Pruning | | | | | | | | | | |
| Training-Aware Pruning Criterion | | | | | | | | | | |
| LLM-Pruner Vec | 28.3 | 47.9 | 62.2 | 73.9 | 59.4 | 60.9 | 64.5 | 35.5 | 35.2 | 55.9 |
| LLM-Pruner E1 | 25.9 | 48.1 | 54.5 | 76.5 | 63.3 | 60.5 | 68.3 | 37.5 | 38.0 | 56.9 |
| LLM-Pruner E2 | 26.3 | 48.6 | 62.3 | 76.4 | 63.3 | 60.1 | 66.7 | 37.3 | 39.1 | 57.8 |
| Inference-Aware Pruning Criterion | | | | | | | | | | |
| Ours (C) w/o remedy | **25.7** | **42.7** | 70.9 | 74.9 | 68.5 | 68.1 | 67.4 | 38.4 | 38.8 | **61.0** |

Table 6: The zero-shot performance of the compressed Vicuna-7B (20% sparsity respective to the global). Following the LLM-Pruner methodology [18], we only prune the transformer blocks from the 4th to the 30th. The average performance is calculated across seven classification datasets.

| Pruning Methods | WikiText2 ↓ | PTB ↓ | BoolQ | PIQA | HellaSwag | WinoGrande | ARC-e | ARC-c | OBQA | Ave ↑ |
|---|---|---|---|---|---|---|---|---|---|---|
| Dense | 16.11 | 61.37 | 76.57 | 77.75 | 70.64 | 67.40 | 65.11 | 41.21 | 40.80 | 62.78 |
| Data Free Pruning | | | | | | | | | | |
| Random | 34.63 | 112.44 | 61.47 | 70.89 | 54.67 | 56.27 | 55.60 | 31.74 | 34.60 | 52.18 |
| L2 norm | 3339.98 | 5882.21 | 55.90 | 56.15 | 32.37 | 51.85 | 30.01 | 28.41 | 28.20 | 40.41 |
| Ours SG w/o remedy | **28.45** | **92.3** | 62.51 | 72.63 | 56.54 | 57.46 | 58.68 | 33.29 | 36.2 | **53.91** |
| Data Dependent Pruning | | | | | | | | | | |
| Training-Aware Pruning Criterion | | | | | | | | | | |
| LLM-Pruner Vec | 27.03 | 92.51 | 62.17 | 71.44 | 55.80 | 53.43 | 55.77 | 33.28 | 37.80 | 52.81 |
| LLM-Pruner E2 | 24.70 | 94.34 | 62.87 | 75.41 | 64.00 | 58.41 | 60.98 | 37.12 | 39.00 | 56.83 |
| LLM-Pruner E1 | 25.74 | 92.88 | 61.70 | 75.30 | 63.75 | 56.20 | 63.22 | 36.60 | 37.00 | 56.25 |
| Inference-Aware Pruning Criterion | | | | | | | | | | |
| Ours (C) w/o remedy | **19.88** | **90.04** | 62.48 | 75.68 | 65.23 | 61.27 | 63.4 | 35.49 | 37.6 | **57.31** |
| Training-Aware Pruning Criterion and Model Retraining | | | | | | | | | | |
| LLM-Pruner LoRA | 18.97 | 76.78 | 60.40 | 75.63 | 65.45 | 63.22 | 63.05 | 37.71 | 39.00 | 57.78 |

# Appendix-F: Novelty and Distinction from Previous Work

To comprehensively address the novelty of our contributions, it is important to revisit the evolution of pruning methods for language models across three distinct stages: the *Pre-Deep Learning Age*, the *Early*

*Deep Learning Age*, and the *LLMs Age*. This historical perspective contextualizes the advancements made by our work and highlights how it differs from prior methods.

**Pre-Deep Learning Age (1970s-1990s):** Pruning research originated in the 1970s through the 1990s, with early studies recognizing that simple magnitude-based pruning often failed to fully preserve network performance due to its neglect of activation effects. In response, researchers explored alternative algorithms based on similarity or variance analysis. One of the most notable works from this era is LeCun's *Optimal Brain Damage* (OBD) from 1989 [71], which utilized second-order derivatives (the Hessian matrix) for pruning. Despite being innovative, this method was computationally expensive for the hardware of that time, confining its application to small-scale networks. These early works laid the foundation for future pruning methodologies, including the ones we build upon in this paper.

**Early Deep Learning Age (2012-2022):** The emergence of deep learning, particularly from 2012 to 2022, brought with it a surge in the development of moderate-sized language models like Transformer, GPT, GPT-2, and BERT. This era saw a significant shift toward applying pruning techniques to larger models, with methods such as the *Pretrained Ticket*, *Lottery Ticket Hypothesis*, *SNIP*, *EBERT*, *BERT-PKD*, *Random Pruning*, and *Movement Pruning* gaining popularity. Many of these methods required multiple rounds of pruning-induced retraining, which became feasible as computational resources improved. The period also witnessed a revival of interest in pruning metrics, including those based on weight magnitude, activations, and higher-order information, many of which had been initially proposed in the previous era.

**LLMs Age (2023-present):** Since 2023, the landscape of language models has drastically evolved with the advent of extremely large models, such as LLaMA, GPT-3, and their successors. The sheer size of these models has rendered traditional retraining-based pruning methods less viable due to prohibitive computational costs. As a result, researchers have increasingly turned their focus toward more efficient pruning strategies that avoid expensive retraining. Prominent works in this era include *LLM-Pruner*, which introduces structured pruning, and *Wanda*, which addresses unstructured and semi-structured pruning. However, these methods still face challenges. For instance, *LLM-Pruner* depends on computationally intensive training-aware metrics and requires at least one round of retraining, while *Wanda* and similar methods, such as *Sheared LLaMA*, necessitate specialized datasets or additional rounds of pretraining. Other approaches like *OPTIN*, *Sliced GPT*, *LLM Surgeon*, *ZipLM*, and *KRP* focus on unstructured pruning or often come with high computational costs due to the use of higher-order information. These limitations point to the growing need for more scalable and retraining-free pruning techniques, which our work addresses.

## Our Novel Contributions

Against this historical backdrop, our work introduces several key innovations that address the limitations of pruning in the age of large language models. We summarize our contribution as follows:

**Optimization Perspective in Pruning.** Firstly, we propose an entirely new framework for pruning large language models from an *optimization perspective*, focusing on *output approximation*. In contrast to prior methods that rely on weight magnitude or gradient-based metrics, we develop pruning criteria based on the output behavior of the model. By prioritizing output approximation, we ensure that the model's performance remains robust, even without the need for costly retraining phases. This conceptual contribution significantly advances the field of structured pruning by emphasizing efficiency and inference-awareness in pruning decisions.

**Identification of Depth-2 Modules.** Additionally, we introduce the novel concept of *depth-2 modules* within transformer-based architectures. We recognize that both the attention mechanism and feed-forward (FFD) layers in transformers share similar structural properties, allowing us to prune them using a unified approach. This insight facilitates the development of a consistent set of pruning metrics and recovery methods that apply across different model components. Our approach departs

from traditional layer-wise pruning techniques by focusing on a more granular and efficient module-wise strategy.

**Inference-Aware and Distribution-Aware Pruning Metrics.** We also propose two new pruning metrics: a *second-moment-based metric* that accounts for the variance in both input data and intermediate activations, and a *similarity-based metric* that evaluates redundancy among attention heads. These metrics offer a more sophisticated pruning approach than conventional magnitude-based methods, ensuring that only the most redundant parameters are removed.

**Enhanced Recovery Method.** Finally, we introduce an *enhanced recovery method* designed specifically for structured pruning of transformer architectures. Unlike traditional recovery methods that operate on a layer-by-layer basis, our recovery technique is adapted to handle the unique challenges of module-wise pruning. By refining the pruned weights within depth-2 modules, we achieve significant performance recovery without requiring additional rounds of retraining, setting our work apart from previous unstructured pruning methods.

In summary, our contributions can be seen as both conceptual and technical. Conceptually, we frame pruning as an optimization problem focused on output approximation, while technically, we introduce depth-2 module pruning, novel pruning metrics, and an efficient recovery process. These advancements represent significant progress in the structured pruning of large language models, particularly in the context of avoiding retraining.

