# OpenReview forum: "Greedy Output Approximation: Towards Efficient Structured Pruning for LLMs Without Retraining"
_CPAL.cc/2025/Proceedings_Track — CPAL 2025 (Proceedings Track) Poster_

### Official Review · Reviewer_pSP9 · 2025-01-04
**Structured pruning for LLMs**

**Rating:** 7
**Confidence:** 3

**Review:**

This paper introduces a method to structuredly prune weights in LLMs and a way to reduce the error induced by pruning. Attention heads are pruned based on a similarity metric to eliminate redundancy while weights in FF layers are pruned at the row/column level.

Pros:
- Easily readable/intuitive figures
- No training required
- Good spread of evaluation tasks/models

Cons
- The motivation for static structured pruning feels lacking. Recent adaptive structured methods [1, 2] are also efficient at inference and training-free.

Questions:
1) One of the motivations for structured pruning is to produce hardware-friendly structures. Do you have measurements on inference efficiency (latency or throughput) with your method?
2) How long (approximately) does it take to prepare a model using your method?
3) To my understanding, attention weights are pruned at the head level. If so, how does your method work for (or have you tried using your method on) models with multiquery or grouped query attention?
4) Have you tried merging similar attention heads together instead of pruning+scaling?
5) The current results still show a significant gap between the full model and the pruned model. Since models usually produce sparser features as they scale in size, the gap may possibly shrink as well. Have you tried some of these tasks on larger models (>10B)?

[1] Lee et al., CATS: Contextually-Aware Thresholding for Sparsity in Large Language Models, 2024.

[2] Dong et al., Prompt-prompted Adaptive Structured Pruning for Efficient LLM Generation, 2024.

---

### Official Review · Reviewer_KwoY · 2025-01-10

**Rating:** 6
**Confidence:** 5

**Review:**

## Summary
This work proposes a zero-order – i.e., gradient free – structured pruning methodology for LLMs that is capable of pruning LLM decoders to up to 20% sparsity by considering the structure of MHA and MLP modules and the channel dependencies between these NN structures with at least two linear layers. Further, this work examines two pruning metrics: 1) Jensen-Shannon divergence for pruning MHA attention heads and, 2) a second-moment based metric for MLPs.

## Strengths
* This work examines and important and timely topic.
* Empirical evaluations include representative downstream evaluation tasks in addition to perplexity on typical datasets.
* The proposed method outperforms the baselines which are relatively robust.
* The proposed method does not require retraining, but rather uses least-squares reconstruction (Algo. 1) to efficiently update the pruned weight matrices without invoking gradient descent. This is a very memory friendly implementation.
* The motivation to remove attention heads based on inter-head similarity is well grounded and appears to work well.

## Weaknesses
* My primary concerns relate to the overall motivation of this work. While I agree with the authors that recovery fine-tuning can be expensive, particularly if full fine-tuning (i.e, not PEFT/LoRA) is required. However, LoRA fine-tuning is actually quite inexpensive for the model sizes examined in this work. For instance, in Table 1 the authors include LLM-Pruner w/ LoRA fine-tuning which appears to perform similar to the proposed method with calibration and least-squares reconstruction. The LoRA fine-tuning step for LLM-Pruner is conducted on a single 4090 in 2.5 hours as outlined in their Appendix B.2, a far cry from being “unaffordable” as described when motivating this work. Fundamentally, real-world impact would require state-of-the-art performance on downstream evaluations in addition to efficient model properties. As such, unfortunately I do not believe this work will be highly impactful or applicable to the vast majority of use cases where model efficiency **and** generalization performance are paramount.
* Given that the proposed method does not require gradient descent to retrain and compresses decoder blocks in a block-wise sequential fashion, I am curious why the authors did not scale their experiments to larger models such as 70B sizes. While I recognize that GPT-2 and LLama 2 7B have been adopted as somewhat standard models to compare with in the structured LLM pruning literature, it seems to me that one of the primary advantages of the proposed method is the apparent ease with scaling to even larger models. I believe showcasing this property would go a long way to improve the potential impact of this work.
* In their discussion of Intra channel pruning, the authors should cite DISP [1], which uses index operators and straight through estimators to completely break the residual dependency issue with intra channel pruning.
* I also note that DISP appears to slightly outperform the proposed method on several of the examined downstream evaluations. I suggest that the authors add some discussion to further contrast their method with DISP (i.e, DISP requires first-order information but also performs well without weight updates, etc.).
* In Appendix F, the authors state that Slice-GPT, ZipLM, and LLM Surgeon focus on unstructured pruning. However, these are all *structured* pruning methods. Please clarify this statement.
* Limited evaluation on math datasets. Including GSM8k or similar math datasets would help establish the cost of structured pruning on that particular dimension of LLM performance.
* I note the sequence length used for measuring PPL was only 128, a very small value and far from the pretraining setting used for these models. While this does allow for consistency when comparing with LLM-Pruner, I note that generally it is best practice to measure PPL using the same context length that the base model was originally trained with (i.e., 4096 for LLaMa 2 7B).

Overall I believe the pros of accepting this paper outweigh the cons as there may be circumstances where recovery fine-tuning is strictly not possible, albeit this circumstance is likely very rare. I would be willing to increase my score if the authors can showcase that their method scales to larger model sizes (~70B) as well where retraining costs are more considerable.

[1]  S. Gao et al., “DISP-LLM: Dimension-Independent Structural Pruning for Large Language Models,” Nov. 04, 2024, arXiv: arXiv:2410.11988. doi: 10.48550/arXiv.2410.11988.

---

### Official Review · Reviewer_gHt5 · 2025-01-14
**A Novel Approach to LLM Pruning Without Retraining**

**Rating:** 6
**Confidence:** 4

**Review:**

**Strengths**

1. The identification and utilization of depth-2 modules in transformer architectures is clever and effective.
2. It introduces an innovative optimization perspective for model pruning, categorizing pruning approaches into three categories, providing a clear theoretical foundation.

**Weaknesses**

1. The authors claims better hardware efficiency, there's limited concrete analysis of actual hardware performance metrics (e.g., speed for the pruning metric calculation, memory bandwidth utilization).
2. How much does each component (similarity-based metric, second-moment-based metric, recovery method) individually contribute to the final performance gains?

---

### Meta-Review · Area_Chair_zzvk · 2025-02-03

**Recommendation:** Accept (Poster)
**Confidence:** 3

**Metareview:**

The submission introduces an innovative zero-order structured pruning method for large language models that notably circumvents the retraining phase. By leveraging a depth-2 pruning structure along with two inference-aware criteria derived from output approximation, the paper contributes a fresh perspective to the pruning literature, especially in the context of efficiently reducing computational overhead and memory usage.

While some concerns were raised regarding the detailed hardware performance metrics and scalability to larger models, the authors have provided comprehensive clarifications that address these issues (with a theoretical analysis of the complexity of their algorithm and an estimation of its runtime in practice). In addition, the method’s ability to maintain competitive performance on various datasets despite a 20% reduction in parameters demonstrates its practical value. The discussion regarding alternative approaches such as LoRA and DISP, along with acknowledgment of the method’s limitations, further strengthens the contribution by situating it appropriately within the current research landscape.

Overall, the work presents a meaningful advance in the area of efficient model pruning for LLMs without retraining, offering both theoretical insights and practical benefits. Given its novel approach and the potential impact on scenarios where retraining is impractical, I believe the paper merits acceptance.

---

### Decision · Program_Chairs · 2025-02-11

Accept (Poster)